# Combined Analysis by GC(RI), GC-MS and 13C NMR of Leaf and Wood Essential Oils from Vietnamese *Glyptostrobus pensilis* (Staunton ex D. Don) K. Koch

Tran Huy Thai [1], Mathieu Paoli [2], Nguyen Thi Hien [1], Nguyen Quang Hung [1], Ange Bighelli [2], Joseph Casanova [2] and Félix Tomi [2,*]

[1] Institute of Ecology and Biological Resources, Vietnam Academy of Science and Technology, Hanoi 10000, Vietnam; thaiiebr@yahoo.com.vn (T.H.T.); nguyenhien333@yahoo.com (N.T.H.); nqhungiebr@yahoo.com (N.Q.H.)

[2] Laboratoire Sciences Pour l'Environnement, Université de Corse-CNRS, UMR 6134 SPE, Equipe Chimie et Biomasse, Route des Sanguinaires, 20000 Ajaccio, France; paoli_m@univ-corse.fr (M.P.); bighelli_a@univ-corse.fr (A.B.); joseph.casanova@wanadoo.fr (J.C.)

* Correspondence: tomi_f@univ-corse.fr

**Abstract:** *Glyptostrobus pensilis* (Staunton ex D. Don) K. Koch is a critically endangered species, native to southeastern China and also very locally found in Dak Lak Province, Vietnam. Essential oil isolated from leaves is a monoterpene-rich oil containing mainly limonene (33.3%), α-pinene (23.4%) and bornyl acetate (9.2%). The composition of *G. pensilis* wood oil is rather complex and the identification of individual components needed fractionation over column chromatography. The main components, identified by GC(RI), GC-MS and 13C NMR, were cedrol (29.3%), occidentalol (6.6%) and occidentalol isomer (5.9%).

**Keywords:** *Glyptostrobus pensilis*; endangered species; wood oil; cedrol; occidentalol





## 1. Introduction

*Glyptostrobus pensilis* (Staunton ex D. Don) K. Koch, also known as Chinese swamp cypress, is the only species in the genus *Glyptostrobus* (Cupressaceae). It is inventoried under various synonyms such as *Glyptostrobus aquaticus*, *G. heterophyllus*, *G. sinensis*, *Sabina aquatica*, *Taxodium japonicum* ssp. *heterophyllum*, *T. sinense*, *Thuja pensilis*. It is native to southeastern China and also very locally found in Dak Lak Province, Vietnam (two natural populations, with over 200 individuals) [1,2].

*G. pensilis* is a medium-sized tree, reaching 20–25 m tall and with a trunk diameter up to 1–1.6 m. Its brown bark is cracked into long, irregular strips. The main branches spread horizontally. The leaves are deciduous, spirally arranged, 5–20 mm long and 1–2 mm broad. The 2–3 cm long and 1–1.5 cm diameter cones are green, going yellow-brown during maturation. They open when mature to release winged seeds, 5–20 mm long. It typically grows in river banks, ponds and swamps [3].

*G. pensilis* is a critically endangered species. Indeed, the species is nearly extinct in the wild due to overcutting for its valuable wood. However, a few specimens are found in several botanical gardens around the world. Genetic variation within and between Chinese populations was investigated using inter-simple sequence repeats (ISSRs). The results show that genetic diversity of *G. pensilis* is rather low [4].

Studies of the last remnants of *G. pensilis* native populations in Vietnam have been carried out including standard taxonomical treatment, ecology, population structure and natural conditions of its habitats [2].

Concerning phytochemicals, a new abietane diterpene, glypensin A, and four known compounds, 12-acetoxy-ent-labda-8(17),13E-dien-15-oic acid, quercetin 3-O-α-L-arabino-furanoside, quercetin 3-O-β-D-galactopyranoside, β-sitosterol, were isolated from the

branches and leaves of *G. pensilis* [5]. More recently, it has been reported that five abietane diterpenes isolated from the branches of *G. pensilis* are potential candidates for the prevention and treatment of SARS-CoV-2 [6]. Six spirobiflavonoid stereoisomers including two new ones, spiropensilisols A and B, were isolated from trunk bark [7]. The composition of leaf waxes has been investigated [8]. The resin of *G. pensilis* consisted mainly of diterpenoids: ferruginol, pisiferol, 6,7-dehydropisiferol, abeo-pisiferol, abeo-carnosol, sugiol, salvinolone and 6-hydroxy-salvinolone. The epicuticular wax extracted from shoots of *G. pensilis* comprised lipids, mainly n-nonacosan-10-ol, n-nonacosan-10-one and *n*-alkanes ranging from C27 to C33, with minor diterpenoids and triterpenoids. No sesquiterpenoids were detected [9].

Concerning volatiles, the composition of *G. pensilis* essential oil was reported [10]. The composition is dominated by $\alpha$-pinene (18.9%) and limonene (23.9%). However, this oil sample was advisedly mentioned as "wood oil" instead of "leaf oil" (personal communication of the authors). This error was corroborated by the recent paper of Schmidt et al. (2016) [11] who reported on sesquiterpene-rich wood oil from *G. pensilis* harvested in Vietnam. Indeed, the composition of that wood oil was dominated by cedrol (16.4%), occidentalol (13.2%) and $\beta$-elemol (8.9%) besides the major component assumed as "dihydro-eudesmol isomer" (18.3%). The odor of this oil sample was defined as "soft woody, slightly terpeny top with fresh and green connotation, later soft woody, fine cedar note, tender warm woody notes reminding of cedar and cypress, later balsamic with slight burning note".

Therefore, the aim of the present study was to investigate the composition of leaf and wood oil samples isolated from the same *G. pensilis* tree from Vietnam.

## 2. Materials and Methods

### 2.1. Plant Materials

Leaves and wood from *G. pensilis* were harvested at Trap Kso, Ea Ho Commune, Krong Nang District, Dak Lak Province, central highlands, Vietnam, in March 2020; geographical coordinates: 12 59′25″ N; 108 17′07″ E, at 713 m above sea level (Figure 1). Plant material was authenticated by Dr. Tran Huy Thai. A voucher specimen was deposited at the herbarium of the Institute of Ecology and Biological Resources (IEBR), Vietnam Academy of Science and Technology (Hanoi) with the reference HN-TN125.

### 2.2. Essential Oil Isolation

Leaves and wood were dried in the shade and hydrodistillated separately, 4–5 days after harvest using a Clevenger apparatus for 3 h for leaves and 4 h for wood, leading to colorless essential oils. Mass of material used for hydrodistillation/volume of collected essential oil: leaves: 3050 g/4.3 mL; wood: 2520 g/5.5 mL. Both oil samples were submitted to GC(RI), GC-MS and $^{13}$C NMR analyses.

### 2.3. GC-FID Analysis

GC-FID analyses were carried out using a Perkin Elmer Clarus 500 (Perkin Elmer, Courtaboeuf, France) system equipped with two FID and two fused-silica capillary columns (50 m length, 0.22 mm i.d., film thickness 0.25 μm), with polydimethylsiloxane (BP-1) and polyethylene glycol (BP-20). The oven temperature was programmed from 60 °C to 220 °C at 2 °C/min and then held isothermal at 220 °C for 20 min; injector temperature: 250 °C; detector temperature: 250 °C; carrier gas: H$_2$ (0.8 mL/min); split: 1/60; injected volume: 0.5 μL. The relative proportions of the essential oil constituents were expressed as percentages obtained by peak-area normalization, without using correcting factors. Retention indices (RIs) were determined relative to the retention times of a series of *n*-alkanes with linear interpolation (Target Compounds (V1.2019) software from Perkin Elmer).

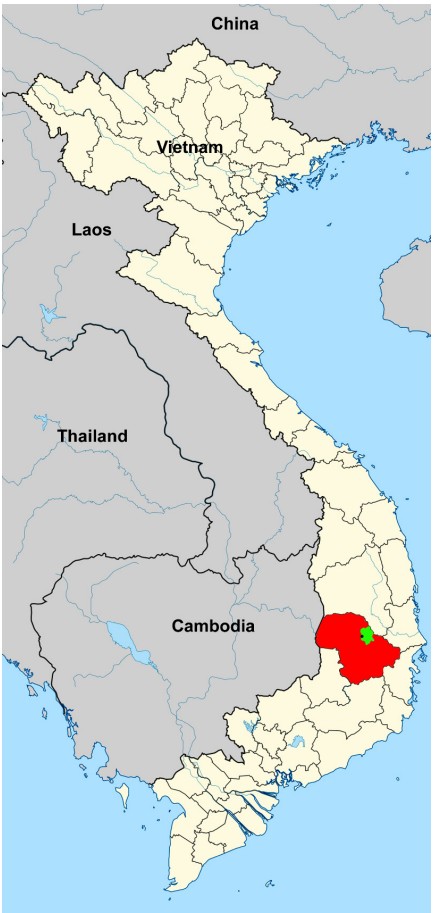

**Figure 1.** Vietnam. Dak Lak Province in red. Krong Nang District in green. Sample collection site is the black dot.

### 2.4. GC/MS Analysis

GC/MS analyses were performed on a Perkin Elmer Clarus SQ8S TurboMass detector (quadrupole), directly coupled to a Perkin-Elmer Clarus 580 Autosystem XL, equipped with a polydimethylsiloxane (BP-1) fused-silica capillary column (50 m length, 0.22 mm i.d., film thickness 0.25 μm). The oven temperature was programmed from 60 to 220 °C at 2°/min and then held isothermal at 220° for 30 min; injector temp., 250 °C; ion source temp., 150 °C; carrier gas, He (1 mL/min); split ratio, 1:80; injection volume, 0.5 μL; ionization energy, 70 eV. The electron ionization (EI) mass spectra were acquired over the mass range 35–350 Da.

### 2.5. Nuclear Magnetic Resonance

[13]C NMR spectra were recorded on a Bruker AVANCE 400 Fourier transform spectrometer operating at 100.63 MHz for [13]C, equipped with a 5 mm probe, in CDCl$_3$, with all shifts referred to internal TMS. The following parameters were used: pulse width = 4 μs (flip angle 45°); acquisition time = 2.7 s for 128 K data table with a spectral width of 25,000 Hz (250 ppm); CPD mode decoupling; digital resolution = 0.183 Hz/pt. The number of accumulated scans was 3000 for each sample (40 mg of essential oil in 0.5 mL of CDCl$_3$).

### 2.6. Identification of Individual Components

Identification of the individual components was carried out: (i) by comparison of their GC retention indices (RIs) on non-polar and polar columns with those of reference compounds compiled in a laboratory-built library and with literature data [12–14]; (ii) by computer matching against commercial mass spectral libraries [14–16]; (iii) by comparison of the signals in the [13]C NMR spectra of the samples with those of reference spectra com-

piled in the laboratory spectral library, with the help of laboratory-made software [17,18]. The usefulness of this technique has been highlighted [19,20], including the identification of epimers and stereoisomers [21,22].

### 2.7. Column Chromatography of the Essential Oil

*G. pensilis* wood oil (1.6803 g) was subjected to column chromatography over $SiO_2$ (250–500 μm, 40 g) and 13 fractions (F1–F13) were eluted using a gradient of solvents (pentane/$Et_2O$, from 0/100 to 100/0) (Table 1). All the fractions of CC have been analyzed by GC(RI) and by $^{13}C$ NMR.

**Table 1.** Components identified by GC(RI) and $^{13}C$ NMR in the 13 fractions of CC (F1–F13), with their relative percentages.

| F1 [a] 100:0 [b] 296 [c] | % | F2 95:5 36 | % | F3 90:10 67 | % | F4 90:10 141 | % | F5 90:10 139 | % | F6 90:10 144 | % | F7 90:10 131 | % |
|---|---|---|---|---|---|---|---|---|---|---|---|---|---|
| 20 | 0.5 | 33 | 3.1 | 52 | 18.1 | 53 | 10.7 | 56 | 85.9 | 56 | 88.7 | 56 | 87.9 |
| 22 | 0.4 | 34 | 5.7 | 53 | 14.2 | 56 | 50.3 | 62 | 2.3 | | | | |
| 23 | 0.9 | 38 | 17.6 | 54 | 18.8 | 62 | 4.4 | 73 | 3.2 | | | | |
| 25 | 0.8 | 39 | 1.7 | 61 | 7.6 | 73 | 6.6 | 74 | 0.4 | | | | |
| 26 | 1.7 | 47 | 2.8 | 65 | 8.6 | 74 | 0.7 | | | | | | |
| 27 | 5.9 | 70 | 17.5 | 66 | 3.9 | 78 | 5.6 | | | | | | |
| 28 | 16.8 | 77 | 1.0 | 70 | 2.0 | | | | | | | | |
| 29 | 1.7 | | | 71 | 3.1 | | | | | | | | |
| 30 | 14.4 | | | 78 | 1.8 | | | | | | | | |
| 31 | 17.2 | | | | | | | | | | | | |
| 32 | 1.4 | | | | | | | | | | | | |
| 35 | 1.0 | | | | | | | | | | | | |
| 36 | 1.3 | | | | | | | | | | | | |
| 37 | 0.7 | | | | | | | | | | | | |
| 41 | 5.6 | | | | | | | | | | | | |
| 43 | 1.0 | | | | | | | | | | | | |
| 44 | 1.6 | | | | | | | | | | | | |
| 45 | 8.5 | | | | | | | | | | | | |
| 77 | 0.9 | | | | | | | | | | | | |
| 79 | 4.8 | | | | | | | | | | | | |

| F8 90:10 162 | % | F9 90:10 111 | % | F10 75/25 149 | % | F11 75:25 107 | % | F12 75:25 99 | % | F13 0:100 95 | % |
|---|---|---|---|---|---|---|---|---|---|---|---|
| 56 | 79.1 | 49 | 5.3 | 49 | 13.8 | 49 | 17.1 | 49 | 14.2 | 49 | 4.8 |
| 60 | 1.7 | 50 | 4.2 | 50 | 15.8 | 50 | 26.6 | 50 | 32.6 | 50 | 15.3 |
| 66 | 0.8 | 55 | 9.0 | 55 | 18.4 | 55 | 16.5 | 55 | 10.6 | 55 | 4.8 |
| 68 | 0.7 | 56 | 47.0 | 63 | 4.1 | 63 | 2.9 | 63 | 1.2 | 56 | 15.9 |
| | | 63 | 3.5 | 66 | 9.5 | 66 | 10.3 | 66 | 7.0 | 67 | 2.1 |
| | | 66 | 3.9 | 68 | 6.9 | 68 | 6.9 | 67 | 2.0 | | |
| | | 68 | 3.0 | | | | | 68 | 4.5 | | |

[a] Fraction number; [b] gradient of pentane:$Et_2O$; [c] mass fraction in mg.

Compound **50**, occidentalol: 139.85 (C, C4), 133.20 (CH, C1), 123.53 (CH, C2), 116.84 (CH, C3), 72.91 (C, C11), 47.66 (CH, C5), 47.26 (CH, C7), 39.09 ($CH_2$, C9), 35.65 (C, C10), 27,30 ($CH_2$, C6), 27.10 ($CH_3$, C12/C13), 26.82 ($CH_3$, C12/13), 26.03 ($CH_3$, C14), 24.80 ($CH_2$, C8), 22.23 ($CH_3$, C15).

Compound **55,** occidentalol isomer: 137.58 (C, C4), 136.62 (CH, C1), 121.28 (CH, C2/C3), 120.77 (CH, C2/C3), 72.92 (C, C11), 45.00 (CH, C5), 44.15 (CH, C7), 34.22 (C, C10), 32.70 ($CH_2$, C9), 27.03 × 2 ($CH_3$, C12/C13), 25.75 ($CH_3$, C14), 24.06 ($CH_2$, C6), 22.18 ($CH_2$, C8), 20.95 ($CH_3$, C15).

## 3. Results and Discussion

Leaf oil and wood oil have been separately isolated using a Clevenger-type apparatus. Yields were 0.143% and 0.220% (*v/w* vs. dry material), respectively.

### 3.1. G. pensilis Leaf Oil

Leaf oil was subjected to GC(RI), GC-MS and $^{13}$C NMR analyses. In total, 28 components have been identified, that accounted for 96.8% of the whole composition (Table 2). The composition is dominated by monoterpene hydrocarbons, limonene (33.3%) and α-pinene (23.4%), followed by camphene (5.6%) and myrcene (4.9%). Bornyl acetate (9.2%) is the major oxygenated monoterpene accompanied by *iso*-bornyl acetate (1.8%). Sesquiterpenes are represented by (*E*)-β-caryophyllene (6.5%) and α-humulene (0.7%) and their oxides, caryophyllene oxide (3.7%) and humulene oxide (0.4%). Abietatriene, (synonym dehydroabietane, 0.5%), 13-epi-pimaradiene (synonym sandaracopimaradiene, 0.7%) and feruginol (0.8%) are the identified diterpenes. These results confirmed that the essential oil previously described by Dai and Thai in 2012 [10] was a leaf oil sample instead of wood oil. Although limonene (23.9%), α-pinene (18.9%), bornyl acetate (5.2%) and (*E*)-β-caryophyllene (6.1%) were its main components, the previously reported oil sample differed from the present one by the content of various components: camphene (5.6 vs. 1.4%), myrcene (4.9 vs. 2.2%), bornyl acetate (9.2 vs. 5.2%). In contrast, the 2012 oil sample contained various sesquiterpenes not detected in the present sample, for instance, among sesquiterpene hydrocarbons, germacrene D (3.6%), β-elemene (2.5%) and β-selinene (2.1%) were found. A few diterpenes, such as ferruginol, were found in both oil samples.

**Table 2.** Chemical composition of leaf and wood essential oils from *Glyptostrobus pensilis*.

| N° | Component | RI Apol Lit | RI Apol | RI Pol | RI Pol Lit | Leaf Oil % | Wood Oil % | Identification |
|---|---|---|---|---|---|---|---|---|
| 1 | Tricyclene | 922 | 920 | 1016 | 1012 | 1.1 | - | RI, MS, NMR |
| 2 | α-Pinene | 934 | 930 | 1016 | 1025 | 23.4 | - | RI, MS, NMR |
| 3 | Camphene | 947 | 943 | 1063 | 1068 | 5.6 | - | RI, MS, NMR |
| 4 | Sabinene | 968 | 964 | 1120 | 1122 | 0.1 | - | RI, MS |
| 5 | β-Pinene | 973 | 969 | 1109 | 1110 | 1.6 | - | RI, MS, NMR |
| 6 | Myrcene | 983 | 980 | 1159 | 1161 | 4.9 | - | RI, MS, NMR |
| 7 | α-Phellandrene | 999 | 996 | 1163 | 1168 | 0.1 | - | RI, MS |
| 8 | α-Terpinene | 1011 | 1008 | 1178 | 1178 | 0.1 | - | RI, MS |
| 9 | p-Cymene | 1015 | 1011 | 1269 | 1270 | 0.1 | - | RI, MS |
| 10 | Limonene * | 1024 | 1022 | 1201 | 1198 | 33.3 | - | RI, MS, NMR |
| 11 | β-Phellandrene * | 1021 | 1022 * | 1209 | 1209 | 0.8 | - | RI, MS, NMR |
| 12 | γ-Terpinene | 1050 | 1047 | 1243 | 1245 | 0.1 | - | RI, MS |
| 13 | Terpinolene | 1079 | 1077 | 1280 | 1282 | 0.2 | - | RI, MS |
| 14 | Linalool | 1086 | 1082 | 1544 | 1543 | 0.1 | - | RI, MS |
| 15 | Borneol | 1153 | 1147 | 1696 | 1700 | 0.1 | - | RI, MS |
| 16 | Terpinen-4-ol | 1164 | 1160 | 1598 | 1601 | 0.1 | - | RI, MS |
| 17 | α-Terpineol | 1176 | 1170 | 1693 | 1694 | 0.4 | - | RI, MS, NMR |
| 18 | Bornyl acetate * | 1270 | 1268 | 1577 | 1579 | 9.2 | - | RI, MS, NMR |
| 19 | *iso*-Bornyl acetate * | 1271 | 1268 | 1582 | 1573 | 1.8 | - | RI, MS, NMR |
| 20 | α-Cubebene | 1352 | 1345 | 1451 | 1460 | - | 0.1 | RI, MS, *NMR* |

**Table 2.** *Cont.*

| N° | Component | RI Apol Lit | RI Apol | RI Pol | RI Pol Lit | Leaf Oil % | Wood Oil % | Identification |
|----|-----------|-------------|---------|--------|------------|------------|------------|----------------|
| 21 | Geranyl acetate | 1361 | 1358 | 1753 | 1751 | 0.6 | - | RI, MS, NMR |
| 22 | α-Copaene | 1375 | 1372 | 1485 | 1491 | - | 0.1 | RI, MS |
| 23 | α-Funebrene | 1385 [a] | 1377 | 1498 | 1500 [a] | - | 0.2 | RI, MS, *NMR* |
| 24 | α-Duprezianene | 1388 [b] | 1381 | 1516 | 1524 [b] | - | 0.2 | RI, MS |
| 25 | β-Elemene | 1388 | 1383 | 1584 | 1591 | - | 0.2 | RI, MS, *NMR* |
| 26 | Sibirene | 1392 [c] | 1396 | 1543 | 1528 [c] | | 0.4 | RI, MS |
| 27 | α-Cedrene * | 1417 [a] | 1408 | 1560 | 1562 [a] | - | 1.4 | RI, MS, NMR |
| 28 | β-Funebrene * | 1415 [a] | 1408 | 1564 | 1570 [a] | - | 3.5 | RI, MS, NMR |
| 29 | β-Cedrene ** | 1425 [a] | 1414 | 1590 | 1594 [a] | - | 2.3 ** | RI, MS, NMR |
| 30 | (*E*)-β-Caryophyllene ** | 1419 | 1414 | 1590 | 1598 | 6.5 | 2.3 ** | RI, MS, NMR |
| 31 | Thujopsene | 1435 [d] | 1424 | 1614 | 1618 [d] | - | 4.4 | RI, MS, NMR |
| 32 | Prezizaene | 1452 [d] | 1440 | 1652 | 1632 [d] | - | 0.4 | RI, MS, *NMR* |
| 33 | (*E*)-β-Farnesene | 1449 | 1445 | 1665 | 1664 | - | 0.3 | RI, MS, *NMR* |
| 34 | α-Humulene | 1449 | 1446 | 1662 | 1667 | 0.7 | 0.3 | RI, MS, NMR |
| 35 | α-Acoradiene | 1462 [e] | 1455 | nd | nd | - | 0.2 | RI, MS, *NMR* |
| 36 | β-Acoradiene | 1462 | 1458 | 1684 | 1688 | - | 0.3 | RI, MS |
| 37 | *trans*-Cadina-1(6),4-diene | 1472 [e] | 1464 | nd | nd | - | 0.4 | RI, MS |
| 38 | ar-Curcumene | 1471 | 1466 | 1773 | 1773 | - | 0.6 | RI, MS, NMR |
| 39 | γ-Curcumene | 1473 | 1468 | 1684 | 1692 | - | 0.5 | RI, MS, NMR |
| 40 | Germacrene D | 1476 | 1481 | 1706 | 1708 | - | 0.2 | RI, MS, *NMR* |
| 41 | Cuparene | 1505 | 1489 | 1815 | 1816 | - | 1.4 | RI, NMR |
| 42 | β-Curcumene | 1503 | 1498 | 1734 | 1737 | - | 0.4 | RI, MS |
| 43 | α-Alaskene | 1512 [e] | 1502 | 1760 | 1763 [f] | - | 0.4 | RI, MS, *NMR* |
| 44 | *trans*-Calamenene | 1513 | 1505 | 1826 | 1823 | - | 0.3 | RI, MS, *NMR* |
| 45 | δ-Cadinene | 1514 | 1510 | 1750 | 1756 | - | 2.4 | RI, MS, NMR |
| 46 | *epi*-Zonarene | 1508 [g] | 1512 | nd | nd | | 0.3 | RI, MS |
| 47 | Zonarene | 1521 [e] | 1519 | nd | nd | - | 0.6 | RI, NMR |
| 48 | α-Calacorene | 1530 | 1523 | 1933 | 1921 | - | 0.1 | RI, MS |
| 49 | β-Elemol | 1537 | 1530 | 2076 | 2088 | - | 4.2 | RI, MS, NMR |
| 50 | Occidentalol | 1548 [h] | 1530 | 2098 | 2097 [h] | - | 6.6 | RI, MS, NMR |
| 51 | (*E*)-Nerolidol | 1550 | 1545 | 2038 | 2036 | - | 0.2 | RI, MS |
| 52 | Caryophyllene oxide | 1570 | 1566 | 1974 | 1986 | 3.7 | 0.9 | RI, MS, NMR |
| 53 | Juniper cedrol | 1583 [d] | 1573 | 2098 | 2102 [d] | - | 2.1 | RI, NMR |
| 54 | Sesquithuriferol * | 1600 [d] | 1582 | 2104 | 2113 [d] | - | 0.9 | RI, NMR |
| 55 | Occidentalol isomer * | nd | 1582 | 2198 | nd | - | 5.9 | RI, NMR |
| 56 | Cedrol | 1597 | 1587 | 2112 | 2120 | - | 29.3 | RI, MS, NMR |
| 57 | β-Biotol | 1595 [d] | 1589 | nd | 2149 [d] | | 0.3 | RI, MS |

**Table 2.** *Cont.*

| N° | Component | RI Apol Lit | RI Apol | RI Pol | RI Pol Lit | Leaf Oil % | Wood Oil % | Identification |
|---|---|---|---|---|---|---|---|---|
| 58 | Humulene oxide | 1597 | 1589 | 2030 | 2047 | 0.4 | - | RI MS |
| 59 | *epi*-Cedrol | 1613 [a] | 1596 | 2166 | 2163 [a] | - | 0.3 | RI, MS, *NMR* |
| 60 | Eudesm-6-en-4α-ol | 1607 [i] | 1606 | 2158 | 2170 [i] | - | 0.7 | RI, *NMR* |
| 61 | 1-*epi*-Cubenol | 1614 | 1610 | 2054 | 2088 | - | 1.1 | RI, MS, NMR |
| 62 | α-Acorenol | 1616 [d] | 1613 | 2128 | 2124 [d] | - | 0.8 | RI, MS, NMR |
| 63 | γ-Eudesmol | 1616 | 1613 | 2161 | 2166 | - | 1.2 | RI, MS, NMR |
| 62 | α-Acorenol | 1616 [d] | 1613 | 2128 | 2124 [d] | - | 0.8 | RI, MS, NMR |
| 64 | τ-Muurolol | 1631 | 1624 | 2186 | 2186 | - | 0.1 | RI, MS, *NMR* |
| 65 | Cubenol | 1620 | 1626 | 2050 | 2068 | - | 0.5 | RI, MS, NMR |
| 66 | β-Eudesmol | 1634 | 1630 | 2222 | 2238 | - | 2.6 | RI, MS-NMR |
| 67 | Selin-11-en-4α-ol | 1640 [i] | 1632 | 2245 | 2249 [i] | - | 0.4 | RI, MS, *NMR* |
| 68 | α-Eudesmol | 1641 | 1635 | 2213 | 2223 | - | 1.8 | RI, MS, NMR |
| 69 | Cedr-8-en-15-ol | 1646 [e] | 1642 | nd | nd | - | 0.6 | RI, MS |
| 70 | Caryophylla-3,8(15)-dien-5β-ol | 1655 | 1649 | 2366 | 2392 | 0.6 | - | RI, NMR |
| 71 | Prezizaan-15-al | 1661 [d] | 1650 | 2142 | 2155 [d] | - | 1.5 | RI, *NMR* |
| 72 | allo-Cedrol (Khusiol) | 1680 [e] | 1659 | 2311 | nd | - | 0.6 | RI NMR |
| 73 | α-Bisabolol * | 1668 | 1662 | 2210 | 2213 | - | 1.3 | RI, MS, NMR |
| 74 | epi-α-Bisabolol * | 1674 | 1662 | 2210 | 2214 | - | 0.2 | RI, MS, *NMR* |
| 75 | β-Acoradienol | 1769 [j] | 1772 | nd | nd | - | 0.2 | RI, MS |
| 76 | 13-epi-Pimaradiene | 1941 [k] | 1954 | 2238 | nd | 0.7 | - | RI, NMR |
| 77 | Abietatriene | 2033 | 2027 | 2483 | 2506 | 0.5 | 0.2 | RI, MS, NMR |
| 78 | Manool | 2047 | 2034 | 2648 | 2628 | - | 0.8 | RI, MS, NMR |
| 79 | Abieta-3,7-diene | 2062 | 2063 | 2444 | 2450 | - | 1.8 | RI, MS, NMR |
| 80 | Feruginol | 2283 [l] | 2273 | nd | nd | 0.8 | 0.2 | RI, NMR |
| | Total identified | | | | | 96.8 | 92.3 | |

Components listed following their elution order on non-polar column (BP-1). Percentages of individual components on non-polar column, except those with *, % on polar column and those with **, % by combination of GC and NMR. NMR: compound identified by [13]C NMR in at least one oil sample; *NMR* (italic) compound identified by [13]C NMR in one or more fraction(s) of CC; RI lit apol and pol ref. [12], otherwise stated: [a] comparison with RIs of pure compounds, [b] [23], [c] [24], [d] [25], [e] [14], [f] [26], [g] [27], [h] [28], [i] [29], [j] [30], [k] [31], [l] [32].

### 3.2. G. pensilis Wood Oil

*G. pensilis* wood oil has also been analyzed by GC(RI), GC-MS and [13]C NMR. In contrast with the leaf oil sample, it is a sesquiterpene-rich essential oil. Indeed, neither monoterpene hydrocarbon nor oxygenated monoterpene has been detected. Various major components have been identified by GC(RI), GC-MS and/or [13]C NMR (Table 2). However, the composition of this oil sample appeared complex, first by the occurrence of uncommon compounds and second by the number of overlapped peaks on the chromatogram that induced, on the one hand, a lack of correspondence between various percentages on non-polar and polar columns, respectively, and, on the other hand, poor fits during GC-MS analysis. Therefore, the sample was subjected to column chromatography (CC) over SiO$_2$ and 13 fractions were eluted using a gradient of solvents (pentane/diethyl oxide, 100/0 to 0/100) and analyzed by GC(RI) and [13]C NMR (Table 1).

Various major components have been identified by combination of the three techniques, GC(RI), GC-MS and $^{13}$C NMR in the whole oil sample. Identification was confirmed by $^{13}$C NMR in one or more fractions of CC. Some minor components have been identified by GC(RI) and GC-MS in the whole oil and by $^{13}$C NMR in fractions of CC. Lastly, a few compounds have been identified in the fractions of CC by $^{13}$C NMR and quantified in the EO through their RIs.

Cedrol (29.3%) significantly dominated the essential oil composition. Among other components bearing the tricyclo[5.3.1.0$^{1,5}$]undecane framework (cedrane), the following have been identified: α-cedrene (1.4%) and β-cedrene (2.3%), as well as β-funebrene (3.5%) and α-funebrene (0.2%, identification ensured by $^{13}$C NMR in fraction B1 of CC). Cedrene and funebrene (α- and β-isomers) differ only by the stereochemistry of the ring junction and $^{13}$C NMR was useful to ensure their identification. Two minor components bearing the cedrane skeleton have been identified, β-biotol (0.3%) and epi-cedrol (0.3%). It could be noted that β-cedrene and (*E*)-β-caryophyllene displayed very close retention indices on non-polar and polar columns and they were fortuitously overlapped on both columns in our oil sample. They have been quantified by combining GC (whole percentage) and $^{13}$C NMR (ratio of the mean intensities of the signals of each component).

Other sesquiterpene hydrocarbons present with appreciable contents were thujopsene (4.4%), δ-cadinene (2.4%) and cuparene (1.4%). Cuparene was identified by GC(RI) and $^{13}$C NMR directly in the essential oil and confirmed in a fraction of CC. Similar identification was performed for zonarene (0.6%). In parallel, calamenene was identified by GC-MS (cis/trans stereochemistry not determined), the trans isomer being differentiated by $^{13}$C NMR in a fraction of CC. Similarly, MS suggested α- or β-alaskene for component 43, the α-isomer being identified by $^{13}$C NMR in fraction F1 of CC.

In addition to the oxygenated cedrane derivatives already mentioned, various oxygenated cyclic or bicyclic sesquiterpenes have been identified in the EO: β-elemol (4.2%), β-eudesmol (2.6%), α-eudesmol (1.8%), γ-eudesmol (1.2%), as well as occidentalol (6.6%). Identification of minor sesquiterpenes demonstrated the utility of combining various techniques. For instance:

- Compound **60** was identified by $^{13}$C NMR in the F8 fraction of CC as eudesm-6-en-4α-ol and quantified in the EO through its retention indices;
- Compound **62** α-acorenol, although co-eluted with γ-eudesmol on the non-polar column, has been identified by MS and $^{13}$C NMR;
- For component **67** (RI apol 1632), MS suggested various sesquiterpenes bearing the bicyclo[4.4.0]decane skeleton and a tertiary alcohol function, such as intermedeol and isomers as well as selina-11-en-4α-ol. The last compound was elected by observation of its chemical shifts in the $^{13}$C NMR spectrum of fractions F12 and F13 of CC.
- Component **71**, prezizaan-15-al, has been identified only by NMR in fraction B2 of CC and quantified in the EO through its retention indices;
- Components **73** and **74** gave overlapped signals on non-polar and polar GC columns and were identified as α-bisabolol or its epimer by GC-MS. $^{13}$C NMR demonstrated the occurrence of both epimers and they were quantified through the ratio of the mean intensities of the corresponding signals.

Special attention should be devoted to components **50** (RIa/RIp = 1530/2098) and **55** (RIa/RIp = 1582/2198), both being identified as occidentalol by GC-MS. The 15 signals of occidentalol were observed in the $^{13}$C NMR spectra of the EO and various fractions of CC and chemical shift values agree with those reported [33]. Component **50** is occidentalol in accordance with RI values, MS spectrum and $^{13}$C NMR data. It accounted for 6.6% in the EO and reached 26.6% in fraction F11. It could be noted that occidentalol was overlapped with β-elemol on the non-polar column used for GC-MS and with juniper cedrol on the polar column.

Occidentalol **50** was identified as early as 1956 in the wood oil from *Thuja occidentalis* [34]. It is a eudesmane derivative characterized by the cis junction of the icy-

clo[4.4.0]decane skeleton and conjugated double bonds. The relative stereochemistry of the substituents has been reported [35,36] and later corrected (Figure 2) [37].

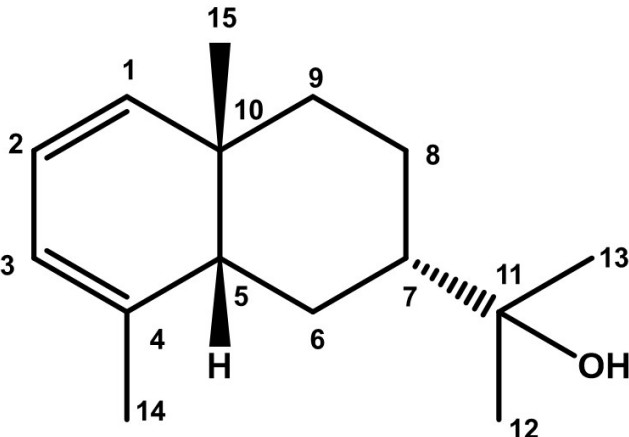

**Figure 2.** Structure of occidentalol **50**.

Although co-eluted with sesquithuriferol on a non-polar column, compound **55** was also identified as occidentalol by MS and both components were differentiated on the polar column (RI = 2104 vs. 2198). It accounted for 5.9% in the whole oil composition and reached 18.4% in fraction B10. Unfortunately, our efforts to purify this compound (successive CC, preparative GC) in order to submit it to a full set of spectroscopic techniques and to elucidate its structure remained unsuccessful, probably due to the great number, in the EO, of oxygenated sesquiterpenes bearing the bicyclo[4.4.0]decane skeleton and a tertiary alcohol function. Therefore, we attempted to assess the structure of **55** with the data at our disposal. Besides **55** (18.4%), fraction B10 contained occidentalol (16%), elemol (14%) and cedrol (9.8%), the $^{13}$C NMR spectra of the three compounds being compiled in our spectral data library. It was possible to extract from the spectrum of B10 the 15 signals belonging to **55**, and to assess the number of hydrogens linked to every carbon through the DEPT spectrum (see Section 2). The molecule contained three quaternary carbons (including an ethylenic carbon and a carbon linked to an oxygen atom), five CH (including three ethylenic carbons), three CH$_2$ and four CH$_3$. These findings corroborated the results of GC-MS that suggested "occidentalol", the MS spectra of **50** and **55** being nearly superimposable. Compound **55** is probably an isomer of occidentalol. Indeed, 7-epi-occidentalol has been reported [38] as well as trans-occidentalol, but $^{13}$C NMR data were not mentioned. Comparing the chemical shifts of occidentalol isomer with those of occidentalol itself, the largest difference is observed for carbon C9, 32.70 ppm vs. 39.09 ppm. The shielding of 6.4 ppm is probably due to a γ steric effect, observable, for instance, in a molecule bearing the cis junction of the bicyclo[4.4.0]decane skeleton and the cis stereochemistry of the methyl and isopropanol groups.

Although some main components of our sample of *G. pensilis* wood oil were previously reported by Schmidt et al. [11], the compositions varied substantially, qualitatively and quantitatively. Indeed, some components displayed higher content in our oil sample, particularly cedrol (29.3% vs. 16.4%). In contrast, other components were less abundant in our oil sample: occidentalol (6.6% vs. 13.2%), elemol (4.2% vs. 8.9%), α-cedrene (1.4% vs. 6.1%). The major component of the Schmidt et al. oil sample, dihydro-eudesmol isomer (18.3%), and dihydro-eudesmol itself (5.7%) were not detected in our sample and, conversely, occidentalol isomer was not suggested in the previous paper. The occurrence of a misprint that reported "dihydro-eudesmol isomer" instead of "dehydro-eudesmol isomer", i.e., occidentalol isomer, could be hypothesized.

According to Hortman et al. [38], "the coincident presence of a rarely occurring cis ring junction and a 1,3-diene system in occidentalol suggests that a unique biosynthetic path-

way". Therefore, we compiled below the plants that produce occidentalol (or occcidentalol isomer) as a secondary metabolite.

Following the pioneering work of Nakatsuka and Hirose [34] who identified occidentalol for the first time (percent not mentioned), a few papers reported on the occurrence of this sesquiterpene in *Thuja occidentalis* wood oil with appreciable content, 19–51%, depending on the duration of hydrodistillation [28,39]. In contrast, occidentalol has never been reported in foliage nor in cone essential oils of this tree [40–44].

However, occidentalol has been identified, to a lesser extent, in essential oils isolated from other species, inter alia, essential oils from wood of *Thuja koraiensis* [45]; from branches of *Taxus canadensis*, 1.4% [46]; from aerial parts of *Bupleurum candollii*, 1.7% [47]; from leaves and fruits of *Julocoton triqueter*, 7.4% [48]; from bark of *Duguetia lanceolata*, 1.4% [49]. Occidentalol has also been observed in tobacco leaves in response to a virus infection [33].

## 4. Conclusions

Leaves and wood of *G. pensilis*, a species scarcely found in Vietnam, in Dak Lak, a province of the central highlands, produced by hydrodistillation essential oils, rich in monoterpenes and sesquiterpenes, respectively. Limonene (33.3%) and α-pinene (23.4%) were the main components of leaf essential oil, which contained also sesquiterpenes and diterpenes to a lesser extent. In contrast, *G. pensilis* wood produced a sesquiterpene-rich essential oil whose composition was significantly dominated by cedrol (29.3%), besides other compounds bearing the tricyclo[5.3.1.0$^{1,5}$]undecane framework (cedrane). Wood oil contained also appreciable contents of occidentalol (6.6%) and occidentalol isomer (5.9%). Yet, none of these compounds can be actually considered as chemotaxonomic markers since they all are widespread compounds in the plant kingdom.

**Author Contributions:** Conceptualization, T.H.T. and J.C.; methodology, N.T.H. and M.P.; formal analysis, M.P. and J.C.; investigation, N.Q.H.; resources, N.Q.H.; writing—original draft preparation, J.C.; writing—review and editing, A.B., F.T. and J.C.; visualization, F.T. and J.C.; supervision, J.C.; project administration, T.H.T. All authors have read and agreed to the published version of the manuscript.

**Funding:** This research was partially funded by Vietnam Academy of Science and Technology, Project code UQĐTCB.05/22-24.

**Data Availability Statement:** The data and $^{13}$C NMR chemical shifts of compounds are available from the authors.

**Acknowledgments:** Vietnamese authors thank the VAST basis investigation project for financial support.

**Conflicts of Interest:** The authors declare no conflict of interest.

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
