# Peer review of "Combined Analysis by GC(RI), GC-MS and 13C NMR of Leaf and Wood Essential Oils from Vietnamese Glyptostrobus pensilis (Staunton ex D. Don) K. Koch"

_compounds, doi:10.3390/compounds3030033_

Round 1

Reviewer 1 Report

The authors have conducted a study on the leaf and wood oils of G. pensilis, confirming the identity of components using 13C NMR along with GCMS.

The article differs from previous work in that a leaf and wood oil from the same plant has been investigated, whereas previously there are only disparate reports of essential oils which might not allow for direct comparison. 

The article is generally written in such a way that meaning is clear and it can be published with minor corrections.

Line 244. This paragraph should be reworded. Include the number 50 referring to the common stereoisomer. Occidentalol has a eudesmane... Rewrite the sentence on stereochemistry/configuration in the paragraph staring line 244

Include 50 in the Figure 1 caption for ease of reference.

Line 194, Remove Obviously,. Why is it obvious that the authors did this? A simple description of the activities taken will suffice.

Line 129. The authors could consider providing their lab made software in some way or as a resource in the data availability section if appropriate. Seems like it might be useful for others. Reword with something like, "with the help of software developed in-house and previously described []."

line 217, rewrite the introduction to this paragraph avoiding the inclusion of Otherwise,

eg, In addition to the oxygenated cedrane.....

Table 1. Consider aligning F8 under F1 in Table 1.

Quality of English is generally good except as noted above.

Reviewer 2 Report

In this paper authors reported the analysis of essential oils from the leaves and wood of Glyptostrobus pensilis by GC-FID, GC-MS, and MNR spectroscopy. In particular, column chromatography was conducted to isolate compounds for identification of essential oils prepared from wood. The following points should be addressed in a revised version of this paper.

 1. The aim of this study was to analyze the constituents of G. pensilis leaf and wood essential oils and compare them to previously reported data. What is the novelty of the present study? I hope that the authors will describe what is different from previous studies for the readers. Furthermore, the authors should state what they expect differences in composition to be based on.

 2. The 13C NMR spectral data for compounds 50 and 55 are reported in this manuscript. I believe that assignment of the 13C NMR spectral data should be conducted.

 3. The authors reported that compound 55 is an isomer of compound 50 in this manuscript. The authors could presume which carbon is stereochemically different based on the 13C NMR spectral data. I hope that the authors should discuss this point.

Round 2

Reviewer 2 Report

I think this manuscript is suitable for publication on this journal.